# Methodological rigor and quality of reporting of clinical trials published with physical activity interventions: A report from the Strengthening the Evidence in Exercise Sciences Initiative (SEES Initiative)

Andresa Conrado Ignacio[3,4], Nórton Luís Oliveira[2,3], Larissa Xavier Neves da Silva[3,4], Jayne Feter[3,4], Angélica Trevisan De Nardi[3,4], Lucas Helal[3,4], Marcelo Rodrigues dos Santos[5], Douglas dos Santos Soares[3,4], Leony Morgana Galliano[1,7], Tainá Silveira Alano[3,6], Daniel Umpierre[1,2,3]*

1 Department of Public Health, Universidade Federal do Rio Grande do Sul, Porto Alegre, RS, Brazil, 2 National Institute of Science and Technology for Health Technology Assessment (IATS/HCPA), Centro de Pesquisa Clínica, Hospital de Clínicas de Porto Alegre, Porto Alegre, RS, Brazil, 3 LADD Lab, Centro de Pesquisa Clínica, Hospital de Clínicas de Porto Alegre, Porto Alegre, RS, Brazil, 4 Postgraduate Program in Health Sciences (Cardiology and Cardiovascular Sciences), Universidade Federal do Rio Grande do Sul, Porto Alegre, RS, Brazil, 5 Faculdade de Medicina, Universidade de São Paulo, São Paulo, Brazil, 6 Medical graduate program, Universidade Federal de Ciências da Saúde de Porto Alegre (UFCSPA), Porto Alegre, RS, Brazil, 7 Universidade Federal do Rio Grande do Norte, Natal, Brazil

* daniel.umpierre@gmail.com

## Abstract

### Background

This study addresses the need for improved transparency and reproducibility in randomized clinical trials (RCTs) within the field of physical activity (PA) interventions. Despite efforts to promote these practices, there is limited evidence on the adherence to established reporting and methodological standards in published RCTs. The research, part of the Strengthening the Evidence in Exercise Sciences Initiative (SEES Initiative) in 2020, assessed the methodological standards and reporting quality of RCTs focusing on PA interventions.

### Methods

RCTs of PA advice or exercise interventions published in 2020 were selected. Monthly searches were conducted on PubMed/MEDLINE targeting six top-tier exercise science journals. Assessments were conducted by two independent authors, based on 44 items originally from CONSORT and TIDieR reporting guidelines. These items were divided into seven domains: transparency, completeness, participants, intervention, rigor methodology, outcomes and critical analysis. Descriptive analysis was performed using absolute and relative frequencies, and exploratory analysis was done by comparing proportions using the $\chi^2$ test ($\alpha = 0.05$).

**Data Availability Statement:** The data for this study are available on the OSF Platform (https://osf.io/wvzcg/).

**Funding:** No specific funding was granted to the SEES Initiative. The project was hosted at the Hospital de Clínicas de Porto Alegre (Porto Alegre, Brazil). ACI received masters degree funding from the Coordenação de Aperfeiçoamento de Pessoal de Nível Superior (CAPES, Brazil - Financial Code 001 - 88887.596408/2020-00). LXNS e JSL received funding for their Ph.D. from the Coordenação de Aperfeiçoamento de Pessoal de Nível Superior (CAPES, Brazil - Financial Code 001 - 42001013017P9). DU receives research productivity fellowship from the Conselho Nacional de Desenvolvimento Científico e Tecnológico (CNPq, Proc 313206/2022-8).

**Competing interests:** The authors have declared that no competing interests exist.

**Abbreviations:** RCT, Randomized Clinical Trials; PA, Physical activity; SEES, Initiative: Strengthening the Evidence in Exercise Sciences Initiative; SRMA, Systematic Reviews with Meta-analysis; CONSORT, Consolidated Standards of Reporting Trials; TIDieR Checklist, Template for Intervention Description and Replication Checklist; PRISMA, Preferred Reporting Items for Systematic Reviews and Meta-Analyses; OSF, Open Science Framework; BJSM, British Journal of Sports Medicine; IJBNPA, International Journal of Behavioral Nutrition and Physical Activity; JSAMS, Journal of Science and Medicine in Sport; MSSE, Medicine and Science in Sports and Exercise; COI, Conflicts of interest; ITT, Intention-to-treat; PP, Per-protocol.

## Results

Out of 1,766 RCTs evaluated for eligibility, 53 were included. The median adherence to recommended items across the studies was 30 (18–44) items in individual assessments. Notably, items demonstrating full adherence were related to intervention description, justification, outcome measurement, effect sizes, and statistical analysis. Conversely, the least reported item pertained to mentioning unplanned modifications during trials, appearing in only 11.3% of studies. Among the 53 RCTs, 67.9% reported having a registration, and these registered studies showed higher adherence to assessed items compared to non-registered ones.

## Conclusions

In summary, while critical analysis aspects were more comprehensively described, aspects associated with transparency, such as protocol registrations/modifications and intervention descriptions, were reported suboptimally. The findings underscore the importance of promoting resources related to reporting quality and transparent research practices for investigators and editors in the exercise sciences discipline.

## Background

In recent years, an increase in the number of studies published in the field of physical activity and health has been observed. As a source of scientific outputs to the clinical context, randomized clinical trials (RCTs) are highly relevant due to their prospective study design and planned interventions, allowing for greater control over potential biases [1, 2]. In the classification of the level of evidence commonly used in clinical guidelines, RCTs are classified as level of evidence B, second only to systematic reviews with meta-analysis (SRMAs), which commonly rely on RCTs as a primary source for their development [3].

Although published studies undergo the peer review process, this process is knowingly vulnerable to flaws resulting from various factors, such as lack of reviewer compensation, lack of specific and standardized training, high inconsistency and lack of agreement among reviewers, as well as various biases related to gender, collaboration, affiliation, geographical location, and race. These factors can influence the reviewer's decision, becoming a risk in the evaluation process, without being based solely on the quality or merit of the work [4, 5]. In view of this, underscoring the importance of post-publication monitoring of methodological and reporting standards.

In the field of physical activity, which encompasses a wide range of techniques and interventions towards the practices of health professionals, it is essential that the evidence maintains a high level of reporting quality. Thus, to enhance the standards of scientific reporting and transparency in RCTs, several guidelines and tools have been developed, with the Consolidated Standards of Reporting Trials (CONSORT) being one of the most disseminated. CONSORT encompasses a set of recommendations and a checklist to guide researchers in reporting RCTs. Derived from CONSORT, the Template for Intervention Description and Replication Checklist (TIDieR Checklist) is also noteworthy, being based on a 12-item checklist that provides guidance for researchers to report on the study interventions [6, 7].

Despite the availability of resources aimed at enhancing transparent scientific reporting, studies indicate that improvements can be made in methodological rigor and completeness in

several health disciplines [8–11]. The identified weaknesses can reduce reproducibility, reliability, and result in diminishing research waste to some extent [12]. Therefore, identifying potential issues in publications can be a key step towards improvement.

With the aim of promoting the monitoring and evaluation of methodological descriptions of studies in the field of physical activity and health, The Strengthening the Evidence in Exercise Sciences (SEES Initiative) was established. By employing established instruments such as CONSORT and TIDieR, this initiative systematically evaluated the methodological and reporting standards of RCTs published in physical activity journals, with the intention of publicly disseminating the results of these analyses [13].

From this, the present study describes the results obtained from the assessments conducted by the SEES Initiative during the 12 months of the year 2020. We evaluated RCTs published in pre-selected journals in the field of physical exercise and quantified the adherence to the recommended methodological and reporting standards for RCTs.

## Methods

This study was written based on the Preferred Reporting Items for Systematic Reviews and Meta-Analyses (PRISMA) statement [14]. This study is of a methodological nature, involving evidence synthesis, and, therefore, did not require submission to the ethics committee or obtaining informed consent from research participants.

### Methodological context

The present study constitutes the main report of results from the monitoring of RCTs conducted by the SEES Initiative. This was a non-profit initiative that continuously assessed publications based on RCTs and SRMAs during the years 2019 and 2020. The purpose of this initiative was to enhance the importance of transparency and reproducibility in research, while adhering to practices of open science.

The detailed methods are available in a protocol manuscript [15]. Briefly, the monthly operationalization was carried out by three committees, namely: (i) pre-evaluation committee, responsible for conducting and managing search strategy on the PubMed (Medline); (ii) evaluation committee, responsible for the analysis of eligibility criteria and data extraction; and (iii) post-evaluation committee, responsible for data analysis, dissemination, storage, and sharing. To ensure proper execution of all processes, the committees followed a manual of operational standards, which is available on the Open Science Framework (OSF) repository (osf.io/ntw7d/).

In this report, the published results refer to the RCTs evaluated from January to December 2020.

### Eligibility criteria

Eligible RCTs were those that included at least one intervention arm of counseling for physical activity or an exercise/physical activity program. Various study designs such as parallel, cluster, non-inferiority, equivalence, factorial, and crossover designs were included in the eligibility criteria. Additionally, secondary clinical trials derived from a primary study were also eligible. The eligibility assessment was conducted by two independent reviewers (AD and LG), and in case of disagreement, a third reviewer (DU) was consulted.

### Search methods

The literature search was conducted monthly between January and December 2020 using the PubMed (Medline) database. A structured and highly sensitive search strategy for retrieving

RCTs was employed [16]. Six journals publishing studies in exercise sciences were included in the searches, as follows: British Journal of Sports Medicine (BJSM), European Journal of Preventive Cardiology, International Journal of Behavioral Nutrition and Physical Activity (IJBNPA), Journal of Science and Medicine in Sport (JSAMS), Medicine and Science in Sports and Exercise (MSSE), and Scandinavian Journal of Medicine & Science in Sports. The complete structure can be found at S1 File.

The selection of these journals was based on criteria related to audience reach, publication frequency, volume of RCTs and/or SRMAs publications, association with scientific societies, and impact factor. Additionally, the selection was based on the potential sample, considering the number of RCTs with physical activity interventions published in these journals in 2019 (osf.io/ntw7d/), it was also observed that the selected journals for 2020 had a higher number of published RCTs in the field of physical activity. All studies identified in the search, along with their eligibility, are available in S2 and S3 Files.

## Screening and data extraction

The screening was performed in duplicate and independently by various team members (AD, LG, DS, MS, AI, TA). This committee assessed the eligibility criteria and, from the eligible studies, conducted the randomization process (via the random.org website) to select five studies to be analyzed each month. The decision to select five studies was based on previous analyses conducted in 2019, taking into consideration the operational capacity of the project team. Due to the burden of activities related to selection, extraction, curation, and dissemination of the analyses, as well as the limited number of researchers involved in the project, it was deemed unfeasible to analyze all eligible studies, resulting in a limit of five randomized clinical trials per month. After study inclusion, data extraction was carried out using an online and standardized form (https://osf.io/wvzcg/),. Subsequently, a data verification process was conducted, and any inconsistencies were resolved through consensus. In cases where discrepancies persisted between the researchers, a third researcher (DU) was consulted for the review of the extracted data. The complete data extraction is available in S4.

## Assessment items and domains

The items used to assess the RCTs were based on (1) what is expected in studies using physical activity interventions or structured exercise programs and (2) the presentation of methodological standards for the purpose of reproducibility, both in clinical practice and in the scientific field. The selection of items for assessment was guided by two guidelines: (a) the CONSORT guideline of 2010 and (b) the TIDieR Checklist (6,7), resulting in a total of 59 items. Only items with dichotomous variables (yes or no) were used for the analysis. Therefore, 14 items that involved polytomous variables (e.g., with options such as incomplete, unclear, or not applicable) were modified to "yes" (meeting recommended practices) or "no" (not meeting recommended practices). These items were evaluated by three researchers (AI, NO, and DU), resulting in a total of 44 items for the final analysis. Further information about these items can be found in the S1 File and in OSF repository (https://osf.io/wvzcg/).

The items have been categorized into seven domains to facilitate the operationalization of the assessment and dissemination. These domains represent broad areas that are significantly impacted when a recommended reporting item is omitted or a methodological routine is not conducted. To avoid redundancy, each item has been assigned to the most relevant domain, even if multiple items could be associated with more than one domain. Below, we describe the creation of the composite domains:

*Transparency Domain*: This domain consists of 5 items, which include study registration, protocol, data sharing plan, description of unplanned modifications to the intervention, and overall changes from the original protocol.

*Completeness Domain*: This domain consists of 10 items, which include description of objectives, study design, brief description of the intervention in the introduction, description of adherence results, recruitment dates, identification of RCT in the title, use of reporting guidelines, funding statement, conflicts of interest (COI) declaration, and study justification.

*Participants Domain*: This domain consists of 2 items, which include inclusion criteria and participant flowchart.

*Intervention Domain*: This domain consists of 9 items, which include intervention description in the abstract, resources used, procedures included in the intervention, professionals involved, delivery mode of the intervention (individual or group, in-person or remote, among others), location of the intervention, period in which it occurred, individual adaptations, and strategies used for adherence.

*Methodological Rigor Domain*: This domain consists of 5 items, which include description of randomization, allocation concealment, blinding, sample size calculation, and use of intention-to-treat (ITT) or per-protocol (PP) analysis strategies.

*Outcomes Domain*: This domain consists of 9 items, which include effect size in the abstract, p-value in the abstract, description of outcomes, measurement method for outcomes, number of participants in the analysis, effect sizes in the results, adverse events, and baseline table.

*Critical Analysis Domain*: This domain consists of four items, which include reporting of hypothesis, statistical analysis, discussion of possible biases, and presence of spin bias. Spin bias is defined as a specific reporting strategy that can distort the interpretation of results and provide incorrect information to readers [17]. This bias occurs when specific reporting strategies are used to emphasize the benefits of the experimental treatment, even if there is no statistically significant difference in the primary outcome, or to distract the reader from non-significant results [18]. In this item, the "yes" option was inverted to "no" to facilitate data analysis, as it represents a non-recommended practice.

These domains were created to facilitate the evaluation process and ensure comprehensive assessment of key aspects of the included studies.

## Dissemination of individual results

The results were disseminated on a monthly basis through three channels: (1) summary reports made available on the SEES Initiative website (www.ufrgs.br/sees-initiative), (2) complete reports were stored on the OSF platform (https://osf.io/ntw7d/), and (3) complete reports were sent to corresponding authors via email. In this case, authors and editors were able to respond and request clarification regarding the reports.

## Statistical analysis

The results are presented using the median and interquartile range, and the variables defined in each domain are reported as absolute frequencies (n) and relative percentages (%).

For most items, the "yes" option was defined as a recommended publishing practice implemented in the RCTs. However, for the count of total recommended practices adhered to in each article, an exception to this rule was the variable of potential spin bias, where the response "yes" indicated a non-recommended practice. Thus, this variable was inverted in the analysis of the frequency of recommended practices in each manuscript to facilitate better understanding of the results (more information in the S1 File).

For the exploratory analysis comparing studies with and without registration, the chi-square test was used. The level of statistical significance was set at $p < 0.05$.

The data extracted were initially stored in a Google Spreadsheet, created from a standardized Google Form, which was filled out in duplicate by the evaluators. The data were analyzed using R software version 4.2.0 (R Development Core Team, Vienna, Austria). The R script and data are openly available in our study repository on OSF and GitHub (osf.io/wvzcg/, github.com/andresaignacio/SeesRcts).

## Results

Out of the 593 studies evaluated, 73 met the eligibility criteria, and, after randomization, 53 were included in the analysis (Fig 1). All included studies, as well as their individual assessment, are available in the S3 File. List of the studies in descending order of adherence to the items/practices recommended and OSF, respectively (https://osf.io/ntw7d/). In the overall analysis of adherence to recommended practices, only one study fully covered all 44 items assessed. In contrast, the study with the lowest adherence covered 18 items. Regarding the seven domains of analysis, the critical evaluation domain, comprising 4 items, had the highest adherence, observed in 35 (66%). Conversely, the outcomes domain, which consists of 9 items,

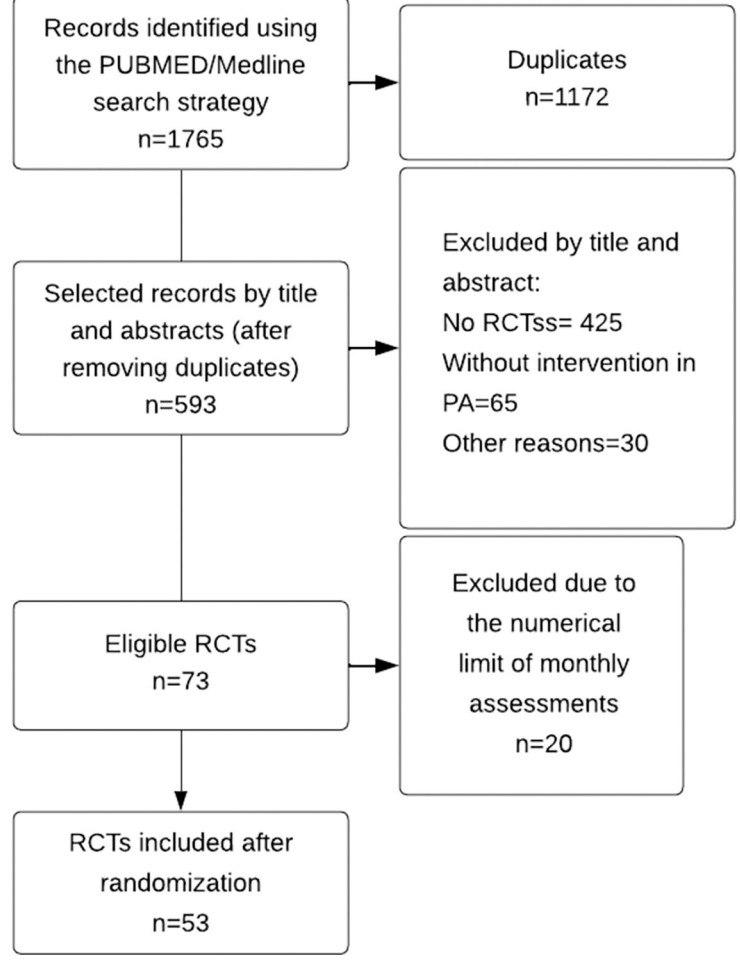

**Fig 1. Flowchart of included RCTs.**

had the lowest reporting adherence, found in only 3 studies (5.7%). The distribution of items within the domains can be seen in Table 1.

### Journal-level reporting

The journal with the most studies assessed was the Medicine & Science in Sports & Exercise. Interestingly, 7 publications in the British Journal of Sports Medicine showed a high item coverage range, with one study displaying all assessed items (Table 2).

### Transparency

For the five items that compose the transparency domain, only 5 (9.4%) studies received a "yes" evaluation on all items, while 12 (22.6%) studies received either no or only one "yes" item, resulting in a reported median of 2 items (0–5).

The item that obtained the most positive evaluation was related to mentioning study registration, with 34 (67.9%) studies covering this item. In contrast, only 6 (11.3%) studies reported modifications to the original protocol, and data sharing was reported by 18 (34.0%) studies.

### Completeness

This domain consisted of 10 items, and 7 (13.2%) studies reported "yes" for all items. On the other hand, the lowest number of items reported as "yes" was 2, which was observed in 4 (3.8%) studies. The median for the quantity of items reported as "yes" in this domain was 7 (5–10).

The most reported items were related to a brief description of the intervention in the study introduction and a description of the rationale in the introduction or methods section 53 (100.0%). On the other hand, the item with the lowest adherence was mentioning the use of guidelines such as CONSORT and/or TIDieR, with only 17 (32.1%) studies reporting it.

### Participants

This domain consisted of two items, both of which 32 (60.4%) of the studies reported. The flow diagram in the results section had the highest number of studies indicating "yes" 41 (77.4%).

### Intervention

This domain comprised a total of 9 items, and only 7 (13.2%) studies received a "yes" evaluation for all items. The majority of studies (30.2%) reported between 7 and 8 items, while the lowest reporting was from 1 study that received a "yes" evaluation for 3 items. The median adherence was 7 items (3–9).

The mode of delivery and the schedule of interventions were the two most reported items in this domain, with almost all studies 52 (98.1%) adhering to the reporting. However, information regarding the description of adherence strategies to the intervention was less reported 12 (22.6%).

### Methodological rigor

This domain consisted of 5 items, with 16 (30.2%) studies covering all of them, and only 3 (5.7%) RCTs not reporting any items. The median of items that received a "yes" evaluation was 3 (0–5).

**Table 1. Frequency distribution of items/recommended practices by domains for the total number of randomized clinical trials (RCTs).**

| Domain | Item | Yes | No |
|---|---|---|---|
| **Transparency** | Register | 36 (67.9%) | 17 (32.1%) |
| | Protocol | 19 (35.8%) | 34 (64.2%) |
| | Data sharing | 18 (34.0%) | 35 (66.0%) |
| | Unplanned modifications regarding the intervention | 6 (11.3%) | 47 (88.7%) |
| | General modifications regarding the protocol | 21 (39.6%) | 32 (60.4%) |
| **Completeness** | Specific objectives | 44 (83.0%) | 9 (17.0%) |
| | Study design | 39 (73.6%) | 14 (26.4%) |
| | A sentence describing the intervention | 53 (100.0%) | - |
| | Results of adherence | 34 (64.2%) | 19 (35.8%) |
| | Recruitment dates | 27 (50.9%) | 26 (49.1%) |
| | Identification of RCT in the title | 23 (43.4%) | 30 (56.6%) |
| | Use of guidelines | 17 (32.1%) | 36 (67.9%) |
| | Statement of sponsors | 49 (92.5%) | 4 (7.5%) |
| | COI* declaration | 49 (92.5%) | 4 (7.5%) |
| | Study justification | 53 (100.0%) | - |
| **Participants** | Eligibility criteria | 39 (73.6%) | 14 (26.4%) |
| | Flowchart | 41 (77.4%) | 12 (22.6%) |
| **Intervention** | Intervention in the abstract | 50 (94.3%) | 3 (5.7%) |
| | Intervention resources | 46 (86.8%) | 7 (13.2%) |
| | Professionals involved | 43 (81.1%) | 10 (18.9%) |
| | Procedures | 50 (94.3%) | 3 (5.7%) |
| | Delivery method | 52 (98.1%) | 1 (1.9%) |
| | Location | 37 (69.8%) | 16 (30.2%) |
| | Schedule | 52 (98.1%) | 1 (1.9%) |
| | Individualization | 38 (71.7%) | 15 (28.3%) |
| | Adherence strategies | 12 (22.6%) | 41 (77.4%) |
| **Methodological rigor** | Randomization | 39 (73.6%) | 14 (26.4%) |
| | Allocation Concealment | 25 (47.2%) | 28 (52.8%) |
| | Blinding | 35 (66.0%) | 18 (34.0%) |
| | Sample size calculation | 38 (71.7%) | 15 (28.3%) |
| | Adherence to ITT** or PP*** | 27 (50.9%) | 26 (49.1%) |
| **Outcomes** | Effect sizes in the abstract | 44 (83.0%) | 9 (17.0%) |
| | P value in the abstract | 36 (67.9%) | 17 (32.1%) |
| | Outcomes in the abstract | 19 (35.8%) | 34 (64.2%) |
| | Outcomes | 39 (73.6%) | 14 (26.4%) |
| | Outcomes measure | 53 (100.0%) | - |
| | Evaluated participants | 40 (75.5%) | 13 (24.5%) |
| | Effect sizes | 53 (100.0%) | - |
| | Adverse events | 18 (34.0%) | 35 (66.0%) |
| | Table baseline | 41 (77.4%) | 12 (22.6%) |
| **Critical analysis** | Hypothesis | 40 (75.5%) | 13 (24.5%) |
| | Statistical analysis | 53 (100.0%) | - |
| | Description of bias | 48 (90.6%) | 5 (9.4%) |
| | *Spin* bias | 5 (9.4%) | 48 (90.6%) |

*RCT: Randomized clinical trial; COI: Conflict of interest; **ITT: Intention-to-treat analysis; ***PP: Per-protocol analysis.

**Table 2. Number of studies per journal and items covered.**

| Journal | RCTs (n) | Items covered per report (min-max range) |
|---|---|---|
| British Journal of Sports Medicine | 7 | 34–44 |
| European Journal of Preventive Cardiology | 2 | 31–33 |
| International Journal of Behavioral Nutrition and Physical Activity | 8 | 35–39 |
| Journal of Science and Medicine in Sport | 5 | 21–28 |
| Medicine & Science in Sports & Exercise | 17 | 23–35 |
| Scandinavian Journal of Medicine & Science in Sports | 14 | 18–39 |
| **Total** | **53** | |

Processes related to participant randomization were reported by 39 (73.6%) studies, contrasting with the adequate description of allocation concealment mechanisms, where 25 (47.2%) studies adhered to the reporting process.

## Outcomes

This domain comprised a total of 9 items, and only 3 (5.7%) studies received a "yes" evaluation for all items, making it the domain with the lowest adherence among the sample. The lowest adherence was from only 1 (1.9%) study, which reported the minimum of 3 items, resulting in a median of 7 (4–8.4) for this domain.

Two items were reported positively by all studies 53 (100.0%), namely the description of outcome measurement methods and the mention of effect sizes in the study results. However, the item least reported was the mention of adverse events 18 (34.0%).

## Critical analysis

Adherence to the 4 items in this domain was 35 (66.0%) studies for all items, and 5 (9.4%) had 2 items evaluated as "yes," making this domain the most addressed among the 53 RCTs, with a median reporting of 4 (2–4) items.

All studies adequately described the statistical analysis in the methods section 53 (100.0%). The item that had the lowest reporting rate was the description of study hypotheses in the introduction 40 (75.5%), but still had good coverage. The remaining items in this domain had an adherence rate above 90%. Spin bias was present in only 5 (9.4%) of the 53 included studies.

## Authors' response

Out of the 53 corresponding authors, only 7 responded to the evaluations sent via email. Among them, five authors requested a revision of at least one item, resulting in a median of contested items of 4 (2–7). One author disagreed with the evaluation but did not request any revisions, while another author fully agreed with the study's evaluation. Only one journal editor responded to the email and did not raise any objections.

## Exploratory analysis

In the exploratory analysis, the adherence to the evaluated items was compared between registered and non-registered studies. Therefore, we used all assessed items except the one relating to registration, totalizing 43 items for comparison. Out of the 53 included studies, 36 (67.9%) had registration or mention of registration in the body of the manuscript. Among the registered studies, one RCT covered all 43 analysis items, whereas among the non-registered

studies, the highest reporting was 32 items. The median of reported items was 33.5 (21–43) in the registered studies and 25 (18–32) in the non-registered ones. Some items showed higher adherence in the registered RCTs, including: protocol (p = 0.005), data sharing (p = 0.001) (transparency domain); recruitment dates (p = 0.0023), trial identification in the title (p<0.001), use of guidelines (p = 0.01), study rationale (p = 0.0090) (completeness domain); eligibility criteria (p = 0.04), flow diagram (p = 0.01) (participants domain); intervention site (p<0.001) (intervention domain); randomization (p<0.001), allocation concealment (p<0.001), blinding (p = 0.02), sample size calculation (p = 0.01), adherence to ITT or PP (p = 0.015) (methodological rigor domain); outcomes (p = 0.006), baseline table (p = 0.01) (outcomes domain); description of biases (p = 0.03) (critical appraisal domain). The results of the 43 items can be visualized in Table 3.

## Discussion

In this study, we evaluated the adherence to reporting standards in RCTs, with a focus on assessing how well these trials followed established guidelines and protocols for reporting their methodologies and findings. The results demonstrated varying degrees of adherence to the recommended methodological and reporting standards as outlined by the CONSORT and TIDieR guidelines. Some items showed high adherence, such as intervention description and statistical analysis, while others showed deficient reporting, such as unplanned modifications and adherence strategies. It is important to note that low adherence to these practices can impact the reproducibility, usability, and accessibility of study results.

Comparable data regarding adherence to these practices have been observed in other areas of the biomedical field, such as pharmacology, pediatrics, and cardiovascular research [19–21]. One of the reasons that may account for these results is the lack of requirement for initial steps like study protocol and registration by journals, in addition to the lack of standardized training for journal reviewers. The degree of adherence to items related to methodological rigor, outcomes' description, and critical evaluation was also variable, which can compromise the interpretation of the quality of evidence from RCTs, as the methodological description is necessary for reproducibility of findings [22]. Furthermore, some studies exhibited spin bias in their results, which can lead the reader to an incorrect interpretation of the actual evidence in the study, as previously observed in RCTs from other fields such as oncology and cardiology [23, 24].

The low adherence to reporting standards in RCTs, lack of standardization in the peer review process, and limitations of scientific journals in accepting studies for publication are relevant issues in the field of scientific research. These problems can have significant consequences for the quality and reliability of results in clinical studies in the field of physical activity. Inadequate reporting of essential methodological details can hinder study replication and impair result interpretation [25]. This can lead to the dissemination of incomplete or biased information, impacting clinical decision-making and health policy formulation [26, 27]. Although the use of reporting guidelines is increasingly endorsed, it does not yet seem to be sufficient to generate proper utilization in the healthcare field at large.

The limitations imposed by scientific journals when selecting studies for publication can inadvertently diminish the quality of reporting. Many journals have requirements related to space constraints and a leaning towards studies that yield statistically significant and impactful results [28]. Therefore, it is necessary to reconsider what is truly essential in publication requests from journals, focusing less on rules such as line count and more on requirements such as study registration and protocol, as well as other factors that endorse transparency and reproducibility.

**Table 3. Frequency distribution of items/recommended practices by domains for registered and unregistered Randomized Clinical Trials (RCTs).**

| Domain | Item | Registered (36) | | Unregistered (17) | | |
|---|---|---|---|---|---|---|
| | | Yes | No | Yes | No | p value |
| Transparency | Protocol | 18 (50.0%) | 18 (50.0%) | 1 (5.9%) | 16 (94.1%) | 0.005* |
| | Data sharing | 18 (50.0%) | 18 (50.0%) | - | 17 (100.0%) | 0.001* |
| | Unplanned modifications regarding the intervention | 5 (13.9%) | 31 (86.1%) | 1 (5.9%) | 16 (94.1%) | 0.65 |
| | General modifications regarding the protocol | 16 (44.4%) | 20 (55.6%) | 5 (29.4%) | 12 (70.6%) | 0.46 |
| Completeness | Specific objectives | 31 (58.5%) | 5 (9.4%) | 13 (76.5%) | 4 (23.5%) | 0.45 |
| | Study design | 27 (75.0%) | 9 (25.0%) | 12 (70.6%) | 5 (29.4%) | 0.75 |
| | A sentence describing the intervention | 36 (100.0%) | - | 17 (100.0%) | - | 0.009 |
| | Results of adherence | 24 (66.7%) | 12 (33.3%) | 10 (58.8%) | 7 (41.2%) | 0.8034 |
| | Recruitment dates | 24 (66.7%) | 12 (33.3%) | 3 (17.6%) | 14 (82.4%) | 0.002* |
| | Identification of RCT in the title | 22 (61.1%) | 14 (38.9%) | 1 (5.9%) | 16 (94.1%) | <0.001* |
| | Use of guidelines | 16 (44.4%) | 20 (55.6%) | 1 (5.9%) | 16 (94.1%) | 0.011* |
| | Statement of sponsors | 33 (91.7%) | 3 (8.3%) | 16 (94.1%) | 1 (5.9%) | 1 |
| | COI* declaration | 35 (97.2%) | 1 (2.8%) | 14 (82.4%) | 3 (17.6%) | 0.092 |
| | Study justification | 36 (100.0%) | - | 17 (100.0%) | - | 0.009 |
| Participants | Eligibility criteria | 30 (83.3%) | 6 (16.7%) | 9 (52.9%) | 8 (47.1%) | 0.042* |
| | Flowchart | 32 (88.9%) | 4 (11.1%) | 9 (52.9%) | 8 (47.1%) | 0.011* |
| Intervention | Intervention in the abstract | 33 (91.7%) | 3 (8.3%) | 17 (100.0%) | - | 0.54 |
| | Intervention resources | 32 (88.9%) | 4 (11.1%) | 14 (82.4%) | 3 (17.6%) | 0.67 |
| | Procedures | 33 (91.7%) | 3 (8.3%) | 17 (100.0%) | - | 0.54 |
| | Professionals involved | 31 (58.5%) | 5 (9.4%) | 12 (70.6%) | 5 (29.4%) | 0.26 |
| | Delivery method | 35 (97.2%) | 1 (2.8%) | 17 (100.0%) | - | 1 |
| | Location | 31 (58.5%) | 5 (9.4%) | 6 (35.3%) | 11 (64.7%) | < 0.001* |
| | Schedule | 35 (97.2%) | 1 (2.8%) | 17 (100.0%) | - | 1 |
| | Individualization | 10 (27.8%) | 26 (72.2%) | 12 (70.6%) | 5 (29.4%) | 1 |
| | Adherence strategies | 10 (27.8%) | 26 (72.2%) | 15 (88.2%) | 2 (11.8%) | 0.30 |
| Methodological rigor | Randomization | 32 (88.9%) | 4 (11.1%) | 7 (41.2%) | 10 (58.8%) | <0.001* |
| | Allocation Concealment | 24 (66.7%) | 12 (33.3%) | 1 (5.9%) | 16 (94.1%) | <0.001* |
| | Blinding | 28 (77.8%) | 8 (22.2%) | 7 (41.2%) | 10 (58.8%) | 0.021* |
| | Sample size calculation | 30 (83.3%) | 6 (16.7%) | 8 (47.1%) | 9 (52.9%) | 0.01* |
| | Adherence to ITT** or PP*** | 23 (63.9%) | 13 (36.1%) | 4 (23.5%) | 13 (76.5%) | 0.014* |
| Outcome | Effect sizes in the abstract | 31 (58.5%) | 5 (9.4%) | 13 (76.5%) | 4 (23.5%) | 0.44 |
| | P value in the abstract | 24 (66.7%) | 12 (33.3%) | 12 (70.6%) | 5 (29.4%) | 1 |
| | Outcomes in the abstract | 14 (38.9%) | 22 (61.1%) | 5 (29.4%) | 12 (70.6%) | 0.72 |
| | Outcomes | 31 (58.5%) | 5 (9.4%) | 8 (47.1%) | 9 (52.9%) | 0.006* |
| | Outcomes measure | 36 (100.0%) | - | 17 (100.0%) | - | 0.009 |
| | Evaluated participants | 27 (75.0%) | 9 (25.0%) | 13 (76.5%) | 4 (23.5%) | 1 |
| | Effect sizes | 36 (100.0%) | - | 17 (100.0%) | - | 0.009 |
| | Adverse events | 23 (63.9%) | 13 (36.1%) | 5 (29.4%) | 12 (70.6%) | 0.87 |
| | Baseline table | 32 (88.9%) | 4 (11.1%) | 9 (52.9%) | 8 (47.1%) | 0.011* |
| Critical analysis | Hypothesis | 26 (72.2%) | 10 (27.8%) | 14 (82.4%) | 3 (17.6%) | 0.51 |
| | Statistical analysis | 36 (100.0%) | - | 17 (100.0%) | - | 0.009 |
| | Description of bias | 35 (97.2%) | 1 (2.8%) | 13 (76.5%) | 4 (23.5%) | 0.032* |
| | S*pin* bias | 34 (94.4%) | 2 (5.6%) | 4 (23.5%) | 13 (76.5%) | 0.076 |

* p<0.05 (significant)

Promoting adherence to and encouragement of appropriate reporting standards in scientific research through the use of guidelines and other educational approaches may foster better practices at the researcher level. However, a collective effort is crucial to improve the scientific culture. In this regard, we underscore the limited engagement of authors and journal editors in accessing evaluations, and similar issues of author receptivity to criticism have been noted in other studies [29, 30].

Typically, the post-publication evaluation tends to focus on journal-related metrics rather than the content of scientific articles. Therefore, initiatives that promote external monitoring are desirable, especially within the field of exercise sciences, where the relevance of interventions have increased alongside the number of study publications. Hence, one of the strengths of our study is its unique aspect in addressing the evaluation of RCTs in physical activity in the post-publication stage. We base our assessment on well-established guidelines and tools, such as CONSORT and TIDieR, providing a strong rationale for our methodological approach. Additionally, we consider the open disclosure of our results to the general public, authors, and editors as a significant strength to foster constructive exchanges and immediate transparency.

Nevertheless, it is important to highlight some limitations of this study. Firstly, we evaluated only RCTs published in the year 2020, and our analysis was conducted on a limited number of journals. Additionally, we analyzed RCTs through sampling. Therefore, caution should be exercised in generalizing the results to RCTs published in other journals. Nonetheless, it is noteworthy that the majority of journals included in the search are affiliated with professional or scientific societies, which can enhance the resources and visibility of the publications. Another limitation is that, although the SEES assessment form was based on well-established guidelines, underwent testing stages, and received input from professionals in the field of physical activity, it did not undergo a formal validation process. Therefore, this factor should be taken into consideration when interpreting the generated data.

In conclusion, our findings highlight low adherence to recommended standards for maintaining methodological rigor and ensuring high-quality reporting in physical activity RCTs. Further efforts seem necessary to align the focus of authors during publication preparation and the criteria used by journals during manuscripts peer review in order to enhance transparency and reproducibility. While certain aspects, such as the interventions' main procedures and interventions' schedule, were adequately addressed, there is considerable room for improvement in areas related to unplanned modification of trials, adherence strategies, and adverse events. Therefore, journals should consider strengthening their review processes, and authors are encouraged to make more extensive use of support tools to provide comprehensive descriptions of their work, ultimately facilitating more effective scientific progress.

## Conclusion

We could observe that adherence to essential criteria for ensuring a high methodological rigor and quality in reports of interventions in randomized clinical trials in the field of physical activity is limited, particularly related to transparency. Additionally, we have identified a concerning deficiency in the acceptance of studies with high levels of methodological rigor and reporting for publication by scientific journals, possibly due to the lack of uniformity in the peer review process. The underutilization of effective tools available to assist researchers and reviewers in the design and evaluation of studies results in the publication of methodologically deficient research in reputable journals, directly compromising the utility of the generated evidence. This scenario not only implies a waste of resources in research but also undermines the fundamental pillars of science.

The findings of this study shed light on the urgent need to establish strict standardization of the peer review process, leading us to question the entire research cycle, from the publication of protocols and records to the post-publication stages. Therefore, it becomes imperative to implement initiatives aimed at closely monitoring evidence after publication, when it is already available to the general public, in order to enhance transparency, communication, and reproducibility in research.

Finally, we emphasize the importance of specialized journals in the field of physical activity to adopt more rigorous standards in the review and acceptance of studies, while authors should make full use of available tools to comprehensively and accurately describe their work. Only through this collaborative approach can we satisfactorily fulfill our shared goal of advancing science and thereby benefiting society as a whole.

## Supporting information

**S1 File.**
(PDF)

**S2 File.**
(PDF)

**S3 File.**
(XLSX)

**S4 File.**
(XLSX)

## Acknowledgments

We would like to express our gratitude to the team involved in the SEES Initiative.

## Author Contributions

**Conceptualization:** Andresa Conrado Ignacio, Nórton Luís Oliveira, Daniel Umpierre.

**Data curation:** Angélica Trevisan De Nardi, Lucas Helal, Marcelo Rodrigues dos Santos, Douglas dos Santos Soares, Leony Morgana Galliano, Daniel Umpierre.

**Formal analysis:** Daniel Umpierre.

**Funding acquisition:** Daniel Umpierre.

**Methodology:** Angélica Trevisan De Nardi, Lucas Helal, Marcelo Rodrigues dos Santos, Douglas dos Santos Soares, Leony Morgana Galliano, Daniel Umpierre.

**Project administration:** Daniel Umpierre.

**Resources:** Angélica Trevisan De Nardi, Lucas Helal, Marcelo Rodrigues dos Santos, Douglas dos Santos Soares, Leony Morgana Galliano.

**Supervision:** Nórton Luís Oliveira, Daniel Umpierre.

**Visualization:** Tainá Silveira Alano.

**Writing – original draft:** Andresa Conrado Ignacio.

**Writing – review & editing:** Andresa Conrado Ignacio, Larissa Xavier Neves da Silva, Jayne Feter, Tainá Silveira Alano, Daniel Umpierre.

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
