## [Decision Letter · Decision Letter 0]

17 Jun 2024

PONE-D-23-40076Methodological Rigor and Quality of Reporting of Clinical Trials Published with Physical Activity Interventions: A Report from the Strengthening the Evidence in Exercise Sciences Initiative (SEES Initiative)PLOS ONE

Dear Dr. Conrado Ignacio,

Thank you for submitting your manuscript to PLOS ONE. After careful consideration, we feel that it has merit but does not fully meet PLOS ONE’s publication criteria as it currently stands. Therefore, we invite you to submit a revised version of the manuscript that addresses the points raised during the review process. The requirement is clarification and design of tables in the manuscript.

We look forward to receiving your revised manuscript.

Kind regards,

Claudia Garcia Serpa Osorio-de-Castro, Ph.D

Academic Editor

PLOS ONE

Additional Editor Comments (if provided):

Reviewers' comments:

Reviewer's Responses to Questions

**Comments to the Author**

1. Is the manuscript technically sound, and do the data support the conclusions?

Reviewer #1: Yes

Reviewer #2: Yes

2. Has the statistical analysis been performed appropriately and rigorously? 

Reviewer #1: Yes

Reviewer #2: Yes

3. Have the authors made all data underlying the findings in their manuscript fully available?

Reviewer #1: Yes

Reviewer #2: Yes

4. Is the manuscript presented in an intelligible fashion and written in standard English?

Reviewer #1: Yes

Reviewer #2: Yes

5. Review Comments to the Author

Reviewer #1: The manuscript is well written. Objectives and methodology are clearly described. Data analysis is adequate and well performed. The results are relevant for evaluating the quality of Clinical Trials submitted for publication, and for producing reliable scientific evidence.

Reviewer #2: This work is innovative and relevant to increasing the reliability of the results of articles in the field. The introduction presents the problem clearly, reviewing the state of the art on the subject. The article presents methods and statistics appropriate to the purpose of answering the experimental question. The results have adequate figures and tables, however, the tables could be better designed. The discussion is pertinent and explains the results found, presenting evidence that leads the reader to reflect and be more careful when absorbing the content of RCT-type articles involving physical activity.

6. PLOS authors have the option to publish the peer review history of their article (what does this mean?). If published, this will include your full peer review and any attached files.

Reviewer #1: **Yes: **Oliveira, Maria Auxiliadora

Reviewer #2: **Yes: **Anderson Luiz Bezerra da Silveira

---

## [Author Response · Author response to Decision Letter 0]

30 Jul 2024

Please ensure that your manuscript meets PLOS ONE's style requirements, including those for file naming:

The manuscript has been reviewed again and adjusted according to PLOS ONE guidelines, as highlighted in the response manuscript.

We note that the grant information you provided in the ‘Funding Information’ and ‘Financial Disclosure’ sections do not match:

The funding information has been reviewed and corrected in the submission.

Please include captions for your Supporting Information files at the end of your manuscript, and update any in-text citations to match accordingly:

The supplementary information has been organized and the corresponding captions have been added in the text and at the end of the manuscript.

Please review your reference list to ensure that it is complete and correct. If you have cited papers that have been retracted, please include the rationale for doing so in the manuscript text, or remove these references and replace them with relevant current references. Any changes to the reference list should be mentioned in the rebuttal letter that accompanies your revised manuscript. If you need to cite a retracted article, indicate the article’s retracted status in the References list and also include a citation and full reference for the retraction notice:

The reference list has been reviewed, and no changes were necessary.

Reviewer Comments to the Author:

Reviewer #1:

"The manuscript is well written. Objectives and methodology are clearly described. Data analysis is adequate and well performed. The results are relevant for evaluating the quality of Clinical Trials submitted for publication and for producing reliable scientific evidence."

Reviewer #2:

"This work is innovative and relevant to increasing the reliability of the results of articles in the field. The introduction presents the problem clearly, reviewing the state of the art on the subject. The article presents methods and statistics appropriate to the purpose of answering the experimental question. The results have adequate figures and tables; however, the tables could be better designed. The discussion is pertinent and explains the results found, presenting evidence that leads the reader to reflect and be more careful when absorbing the content of RCT-type articles involving physical activity."

Tables 1 and 3 have been reorganized to facilitate visualization. All changes made have been duly highlighted in the article.

---

## [Editor Report · Decision Letter 1]

6 Aug 2024

Methodological Rigor and Quality of Reporting of Clinical Trials Published with Physical Activity Interventions: A Report from the Strengthening the Evidence in Exercise Sciences Initiative (SEES Initiative)

PONE-D-23-40076R1

Dear Dr. Ignacio,

We’re pleased to inform you that your manuscript has been judged scientifically suitable for publication and will be formally accepted for publication once it meets all outstanding technical requirements.

Kind regards,

Claudia Garcia Serpa Osorio-de-Castro, Ph.D

Academic Editor

PLOS ONE
---

## [Editor Report · Acceptance letter]

20 Aug 2024

PONE-D-23-40076R1 

PLOS ONE

Dear Dr. Conrado Ignacio, 

I'm pleased to inform you that your manuscript has been deemed suitable for publication in PLOS ONE. Congratulations! Your manuscript is now being handed over to our production team.

Kind regards, 

on behalf of

Dr. Claudia Garcia Serpa Osorio-de-Castro 

Academic Editor

PLOS ONE